# Cyclic AMP Regulates Key Features of Macrophages via PKA: Recruitment, Reprogramming and Efferocytosis

**DOI:** 10.3390/cells9010128

**Published:** 2020-01-06

**Authors:** Graziele L. Negreiros-Lima, Kátia M. Lima, Isabella Z. Moreira, Bruna Lorrayne O. Jardim, Juliana P. Vago, Izabela Galvão, Lívia Cristina R. Teixeira, Vanessa Pinho, Mauro M. Teixeira, Michelle A. Sugimoto, Lirlândia P. Sousa

**Affiliations:** 1Departamento de Análises Clínicas e Toxicológicas, Faculdade de Farmácia, Universidade Federal de Minas Gerais, Belo Horizonte 31270-901, Brazil; grazieleleticia17@gmail.com (G.L.N.-L.); bella_zm@hotmail.com (I.Z.M.); gardenbru@hotmail.com (B.L.O.J.); liviacrteixeira@gmail.com (L.C.R.T.); 2Programa de Pós-Graduação em Ciências Farmacêuticas, Faculdade de Farmácia, Universidade Federal de Minas Gerais, Belo Horizonte 31270-901, Brazil; katiamaciellima@yahoo.com.br; 3Departamento de Bioquímica e Imunologia, Instituto de Ciências Biológicas, Universidade Federal de Minas Gerais, Belo Horizonte 31270-901, Brazil; julypri@gmail.com (J.P.V.); mmtex.ufmg@gmail.com (M.M.T.); 4Departamento de Morfologia, Instituto de Ciências Biológicas, Universidade Federal de Minas Gerais, Belo Horizonte 31270-901, Brazil; izabelag@gmail.com (I.G.); vpinhos@gmail.com (V.P.); 5Programa de Pós-Graduação em Doenças Infecciosas e Medicina Tropical, Escola de Medicina, Universidade Federal de Minas Gerais, Belo Horizonte 30130-100, Brazil; michellesugimoto@gmail.com

**Keywords:** inflammation resolution, db-cAMP, macrophage recruitment, macrophage reprogramming, efferocytosis

## Abstract

Macrophages are central to inflammation resolution, an active process aimed at restoring tissue homeostasis following an inflammatory response. Here, the effects of db-cAMP on macrophage phenotype and function were investigated. Injection of db-cAMP into the pleural cavity of mice induced monocytes recruitment in a manner dependent on PKA and CCR2/CCL2 pathways. Furthermore, db-cAMP promoted reprogramming of bone-marrow-derived macrophages to a M2 phenotype as seen by increased Arg-1/CD206/Ym-1 expression and IL-10 levels (M2 markers). Db-cAMP also showed a synergistic effect with IL-4 in inducing STAT-3 phosphorylation and Arg-1 expression. Importantly, db-cAMP prevented IFN-γ/LPS-induced macrophage polarization to M1-like as shown by increased Arg-1 associated to lower levels of M1 cytokines (TNF-α/IL-6) and p-STAT1. In vivo, db-cAMP reduced the number of M1 macrophages induced by LPS injection without changes in M2 and Mres numbers. Moreover, db-cAMP enhanced efferocytosis of apoptotic neutrophils in a PKA-dependent manner and increased the expression of Annexin A1 and CD36, two molecules associated with efferocytosis. Finally, inhibition of endogenous PKA during LPS-induced pleurisy impaired the physiological resolution of inflammation. Taken together, the results suggest that cAMP is involved in the major functions of macrophages, such as nonphlogistic recruitment, reprogramming and efferocytosis, all key processes for inflammation resolution.

## 1. Introduction

Inflammation is a complex host response against invading pathogens or following sterile tissue injury. The inflammatory response is a spatially and temporally orchestrated process, in which cells and mediators collaborate to neutralize and eliminate the damaging stimuli to allow maintenance of homeostasis. Inflammation can be divided into three main phases: onset, resolution and the more recently described post-resolution phase [1,2,3,4]. If successful, the inflammatory response tends to progress from the onset to the post-resolution phase through a coordinated series of molecular and cellular events that lead to the restoration of tissue structure, organ function, and ‘adapted homeostasis’ [5]. A failed or impaired resolution may underpin the pathogenesis of various chronic inflammatory or autoimmune diseases, such as rheumatoid arthritis, asthma and multiple sclerosis [3,4,6]. Ideally, the inflammatory response protects the host, is self-limiting and progresses to complete resolution [7].

The resolution of inflammation is an active process involving biochemical mediators and signaling pathways controlling switching from pro-inflammatory mediators generation to production of pro-resolving molecules, leukocyte apoptosis (especially of neutrophils) followed by efferocytosis, and switching from pro-inflammatory to pro-resolving cell phenotypes (especially relevant to macrophages) [6,7]. Clearance of apoptotic cells by efferocytosis is an important event that brings about resolution of inflammation and induces remarkable macrophage phenotypic and functional changes [8]. Several studies have shown the ability of macrophages to adopt different phenotypes during an inflammatory process [9]. In brief, M1 macrophages (classically activated macrophages) are involved in the beginning of inflammation and enable host defense against infection, whereas M2 (alternatively activated macrophages) and proresolving (Mres) macrophages have anti-inflammatory and tissue remodeling properties, playing key roles on resolution of inflammation [10,11,12]. The unique ability of macrophages to respond to different types of agonists and to exhibit distinct functions is accompanied by changes in thousands of different genes, transcription factors, cell surface markers and cytokines [13]. This process is referred to as macrophage reprogramming or repolarization [14,15]. Responding to different microenvironments, primary macrophages (M0) can be polarized toward pro-inflammatory (M1) or anti-inflammatory (M2) phenotypes [16]. M1-like macrophages can be re-educated to M2 and consequently to the Mres phenotypes by various effector molecules or following the uptake of apoptotic cells [10,17,18,19,20]. Conversion from M2 to M1 has also been reported, a feature especially relevant to cancer research [21,22]. Macrophage polarization and reprogramming play important roles in the maintenance of immune system homeostasis [23] and provide targets for treating diseases characterized by abnormalities in macrophage function and activation status [24,25], such as cancer and atherosclerosis.

cAMP is a second intracellular messenger produced by adenylate cyclase (AC) from ATP, and its levels are regulated by phosphodiesterases (PDEs) [26,27]. cAMP has been identified as an important intracellular mediator impacting the performance of macrophage functions such as phagocytosis [28] and reprogramming [29,30,31]. Evidence also suggest that elevation of cAMP is an important intracellular event for inflammation resolution [29,32,33,34]. Studies have shown that pro-resolving mediators such as resolvin (Rv) D2 [35], RvD1 [36], melanocortin peptides [37,38] and Annexin A1 (AnxA1) [39] can have their activities mediated by cAMP activation. We have previously shown that cAMP elevating agents up-regulate AnxA1 in vivo and in cultured macrophages, and that AnxA1, in turn, is involved in the pro-resolving actions of such compounds [32]. Since AnxA1 is a molecule endowed with several pro-resolving properties [3,39,40], cAMP seems to be a link between different pathways, widening the resolution cascade [3,32]. Noteworthy, the use of selective phosphodiesterase 4 (PDE4) inhibitors (which promote increased intracellular cAMP levels) for the treatment of predominantly inflammatory diseases, such as Alzheimer, psoriasis, arthritis, chronic obstructive pulmonary disease (COPD), and asthma has been widely studied [41,42,43], and roflumilast, an active oral PDE4 inhibitor, was approved by the FDA aiming to reduce COPD exacerbations [43]. Although the importance of cAMP in the resolution of inflammation is evident, the mechanisms by which cAMP acts in the resolution process are still not completely elucidated.

In the present study, we have investigated the effect of db-cAMP, a cAMP mimetic, on key functions of macrophages in the context of inflammation resolution. Our data suggest that db-cAMP mediates monocyte recruitment in a CCR2/CCL2- and PKA-dependent manner. Furthermore, db-cAMP favors an M2-like macrophage phenotype, either alone or in the presence of IFN-γ/LPS stimuli and amplifies the IL-4-induced M2 phenotype. It also increases the frequency of macrophages expressing engulfment molecules and the efferocytosis of apoptotic neutrophils both in vitro and in vivo. Finally, inhibition of the cAMP effector protein PKA, in the course of acute pleurisy induced by LPS impairs self-resolving inflammation. Our results support that cAMP is involved in the major functions of macrophages such as nonphlogistic recruitment, reprogramming and efferocytosis, all key processes for inflammation resolution.

## 2. Materials and Methods

### 2.1. Animals

All described procedures had prior approval from the Animal Ethics Committee of Universidade Federal de Minas Gerais (CEUA/UFMG, protocol number 183/2017) and Research Ethics Committee of Universidade Federal de Minas Gerais (COEP/UFMG, protocol number 0319020300-11). Male BALB/c mice (8–10 weeks) obtained from the Center of Bioterism of Universidade Federal de Minas Gerais (Belo Horizonte, Brazil) had free access to food and water (ad libitum) and were housed under standard conditions of humidity (50–60%), light (12 h/12 h light/dark cycle) and temperature (22 ± 1 °C).

### 2.2. Drugs, Reagents, and Antibodies

Db-cAMP, H89, staurosporine, CFSE and LPS (from *Escherichia coli* serotype O:111:B4) were from Sigma-Aldrich (San Luis, MO, USA); IFN-γ and IL-4 were from Biolegend (San Diego, CA, USA); RS504393 (Tocris, Bristol, England, UK); western blot antibodies were from Sigma (β-actin), Cell Signaling Technology (Danvers, MA, USA; STAT1, p-STAT1, p-STAT3, secondary anti-rabbit peroxidase conjugate antibody) or Santa Cruz Biotechnology (Dallas, TX, USA; secondary anti-mouse peroxidase conjugate antibody); ELISA kits for measurement of IL-10, TGF-β, CCL2, IL-6 and TNF-α were from R&D Systems (Minneapolis, MN, USA). The fluorescent monoclonal antibodies were anti-F4/80 (PE-Cy7 or APC, eBioscience, San Diego, CA, USA), anti-GR1 (PE, eBioscience), anti-CD11b (alexa fluor 488, Biolegend, San Diego, CA, USA and V500, Pharmingen), anti-rabbit secondary (Alexa fluor 488 Cell Signaling, Danvers, MA, USA), anti-AnxA1 (Santa Cruz Biotechnology), anti-Ly6C (PeCy7, Biolegend), anti-Ly6G (APCCy7 or BV421, Biolegend), anti-CD36 (APC, BD biosciences) and anti-CD3 (FITC, Pharmingen).

### 2.3. Leukocyte Migration to the Pleural Cavity Induced by db-cAMP

Mice were injected intrapleurally (i.pl.) with db-cAMP (4 mg/kg) or PBS. Cells in the pleural cavity were harvested 4, 24 and 48 h after db-cAMP injection by washing the cavity with 2 mL of PBS. In another protocol, mice were pre-treated with specific inhibitors H89 (4 mg/kg, i.pl.) or RS504393 (2 mg/kg, i.pl.) 1h before db-cAMP injection. Cells in the pleural cavity were harvested 48 h after db-cAMP injection by washing the cavity with 2 mL PBS. Total cell counts were determined using Turk’s stain in a modified Neubauer chamber. Differential cell counting was performed using standard morphological criteria to identify cell types on cyto-centrifuge preparations (Shandon Elliott) stained with May-Grünwald-Giemsa. The results are presented as the number of cells per cavity. For a deep investigation of the leukocyte population recruited after db-cAMP, pleural cells were recovered 48 h after db-cAMP or PBS injection and analyzed by flow cytometry using labeling for different leukocyte populations: macrophages (F4/80^+^), monocytes (Ly6C^+^ F4/80^−^), neutrophils (Ly6G^+^) and lymphocytes (CD3^+^). The results are presented as the mean percentage of cells per cavity.

### 2.4. LPS-Induced Pleurisy Model and Treatment with db-cAMP or Inihibition of PKA Using H89

Animals received an i.pl. injection of LPS (250 ng/cavity) or PBS as previously described [32,44] and 8 h later (at the peak of inflammation) were treated with db-cAMP (4 mg/Kg, i.pl.). Cells recruited to the pleural cavity were recovered 30 h following LPS challenge or PBS injection by washing the cavity with 2 mL of PBS. Total cell counts were determined using Turk’s stain in a modified Neubauer chamber. The number of macrophages was assessed by flow cytometry using antibodies to identify three macrophages subpopulations: M1 (F4/80^low^ Gr1^+^ Cd11b^med^), M2 (F4/80^high^ Gr1^−^ Cd11b^high^) and Mres (F4/80^med^ Cd11b^low^), as previously described [12,44,45,46]. In addition, the frequency of macrophages positive for AnxA1 and CD36, important molecules for efferocytosis, was verified by flow cytometry (FACS Canto II, BD biosciences). These results are presented as the mean number or frequency of cells per cavity.

In another protocol, mice were challenged with LPS (250 ng/cavity) or PBS and further injected with H89 (4 mg/kg, i.pl.) at the peak of inflammation [44]. Cells recruited to the pleural cavity were recovered 24 h following LPS challenge or PBS injection by washing the cavity with 2 mL of PBS. To verify the effect of cAMP inhibition on the spontaneous resolution of LPS-induced pleurisy and to calculate the resolution indices [32,44,47], LPS-challenge mice were injected with H89 (4 mg/kg, i.pl) at 8 h and 24 h (booster dose) after LPS. Cells recruited to the pleural cavity were recovered at 48 h following LPS challenge or PBS injection by washing the cavity with 2 mL of PBS. Total cell counts were determined using Turk’s stain in a modified Neubauer chamber. Differential cell counting was performed using standard morphological criteria to identify cell types on cyto-centrifuge preparations (Shandon Elliott) stained with May-Grünwald-Giemsa. The results are presented as the number of cells per cavity. Resolution indices were calculated as described [32,48] as: (i) magnitude (Ψ_max_—the maximum PMN numbers in the exudates) and T_max_ (time point when PMN numbers reach maximum); (ii) duration (T_50_—time point when PMN numbers reduce to 50% of maximum); and (iii) resolution interval R*_i_* (the time period when 50% PMNs are lost from the pleural cavity; i.e., T_50_ − T_max_).

### 2.5. In Vivo Efferocytosis Assay

This protocol is an adaptation of previously described protocols [37,49] and it has been applied in previous publication of our group [45,46,50]. Mice received an intraperitoneal (i.p.) injection of zymosan 0.1 mg/cavity. After 62 h, the animals were treated with db-cAMP (4 mg/kg, i.p.) for 7 h and after received an intraperitoneal injection (i.p.) of 10^6^ apoptotic human neutrophils labeled with carboxyfluorescein diacetate succinimidyl ester (CFSE). Neutrophils were isolated from peripheral blood of healthy human donors, using Histopaque 11,191 and 10,771 (Sigma-Aldrich) as previously described [48]. Isolated neutrophils were stimulated with staurosporine (10 μM, 1h) for induction of apoptosis and labeled with 10 μM CFSE for 1 h at 37 °C under light protection. The percentage of apoptosis was determined in cytospin preparations, counted using oil immersion microscopy (100× objective) to determine the proportion of cells with high distinctive apoptotic morphology [32,34,48,51] and >90% were apoptotic. One hour after injection of the apoptotic neutrophils, cells from the peritoneal cavity were collected with 4 mL of PBS, labeled with fluorescent anti-F4/80 and analyzed by flow cytometry (FACS Canto II, BD Biosciences). Moreover, cell counts were performed on cyto-centrifuge preparations (Shandon III) stained with May-Grunwald-Giemsa to determine the percentage of cells with efferocytic morphology (macrophage with apoptotic bodies observed in their cytoplasm) [45,46]. The results of flow cytometry are presented as percentage of F4/80^+^/CFSE^+^ cells and mean florescence intensity (MFI) of CFSE (flow cytometer laser set at 488) or by morphological analysis by determining the proportion of macrophages that ingested apoptotic neutrophils (500 cells/slides were counted).

### 2.6. Isolation and Culture of Murine Bone Marrow-Derived Macrophages (BMDMs)

BMDMs were prepared as previously described [46,52] with modifications. Bone marrow was collected from tibias and femurs of BALB/c mice and washed with RPMI (Cultilab, Campinas, São Paulo, Brazil). The suspension obtained was then centrifuged for 5 min at 1200× *g*. The pellet of cells was resuspended with complete conditioned media for BMDM differentiation [RPMI with 20% (*v*/*v*) heat-inactivated fetal bovine serum (FBS) and 30% (*v*/*v*) L929 cell-conditioned medium (LCCM)], seeded on tissue culture plates, and incubated at 37 °C with 5% CO_2_. After 3 days, the medium was supplemented with additional complete conditioned media. At day 7 the supernatant was removed, and adherent macrophages were detached using a cell scraper and plated for the different experiments, as described below. Cell differentiation and purity was determined by flow cytometry using the anti-F4/80, marker for macrophages. To induce macrophage polarization to M1-like or M2-like phenotypes, BMDMs were treated with different mouse recombinant proteins as follow: IFN-γ (10 ng/mL) + LPS (10 ng/mL) to induce M1 macrophages or IL-4 (20 ng/mL) to induce M2 macrophages.

### 2.7. RAW264.7 Cell Culture

RAW264.7 murine macrophages were obtained from the American Type Culture Collection (ATCC, Manassas, VA, USA). The cell cultures were maintained in DMEM (Cultilab) supplemented with 10% (*v/v*) FBS at 37 °C in 5% CO_2_. After reaching 70–80% confluence, cells were serum-deprived overnight, and cell viability was determined using a trypan blue dye exclusion assay. To induce macrophage polarization, RAW264.7 cells were treated with different mouse recombinant proteins as follow: IFN-γ (10 ng/mL) + LPS (10 ng/mL) to induce M1 macrophages or IL-4 (20 ng/mL) to induce M2 macrophages.

### 2.8. In Vitro Efferocytosis Assay

In vitro efferocytosis assay was performed as previously described [46], by co-culturing BMDMs with human apoptotic neutrophils labeled with CFSE in a proportion of three apoptotic neutrophils: one macrophage for 1 h. Neutrophils that had not been phagocytosed were removed by vigorous washing of the wells with PBS three times. Efferocytosis by adherent macrophages was assessed by flow cytometry (FACS Canto II, BD biosciences), and the results were expressed as mean florescence intensity (MFI) of CFSE in the F4/80^+^ macrophages (flow cytometer laser set at 488).

### 2.9. Western Blotting

Whole-cell extracts were prepared as described [32,44]. The protein content of the lysate was determined by Bradford assay reagent (Bio-Rad, Hercules, CA, USA). Cell lysates (40 µg) were electrophoresed on denaturing 8–10% polyacrylamide SDS gels under reducing conditions and electrotransferred to nitrocellulose membranes. Membranes were blocked for 1 h with PBS containing 5% (*w*/*v*) nonfat dry milk and PBS containing 0.1% Tween-20 (Synth), washed three times and then incubated overnight with specific primary antibodies for STAT1(1:1000), p-STAT3 (1:1000), p-STAT1 (1:1000) or β-actin (1:10000) in PBS containing 5% (*w/v*) BSA and 0.1% Tween-20. After washing, the membranes were incubated with appropriate horseradish peroxidase-conjugated antibody (1:3000). Immunoreactive bands were Chicago, IL, USA). For densitometry analysis, membranes were scanned, and the bands were quantified using ImageJ software (ImageJ, National Institutes of Health, Bethesda, MD, USA). The results were expressed as arbitrary units and normalized to the values of β-actin or STAT1 in the same sample.

### 2.10. qPCR Analysis of M1 and M2 Macrophage Markers

Total RNA from BMDMs was extracted using the TRIzol™ Reagent (Invitrogen, Carlsbad, CA, USA) according to the manufacturer’s instructions. The cDNA was synthesized using 1 µg of RNA with the SuperScript III Reverse Transcriptase (Invitrogen), according to the manufacturer’s instructions. Real-time PCR was performed in duplicate, with obtained cDNA, specific primers and Power SYBR Green PCR Master Mix (Applied Biosystems, Foster City, CA, USA), using the StepOne™ System (Applied Biosystems). The data were analyzed using StepOne™ System software with a cycle threshold (Ct) in the linear range of amplification and then processed by the 2^–ΔΔCt^ method. The dissociation step was always included to confirm the absence of unspecific products. GAPDH was used as an endogenous control to normalize the variability in expression levels and results were expressed as fold increase. Expression levels for Arginase-1, mannose-receptor (CD206) and iNOS were calculated. Primers (IDT) used were as follows: iNOS (5′-AGCACTTTGGGTGACCACCAGGA-3; 5′-AGCTAAGTATTAGAGCGGCGGCA-3′), Arginase-1 (5′-TGACATCAACACTCCCCTGACAAC-3′; 5′-GCCTTTTCTTCCTTCCCAGCAG-3′), Mannose-receptor-CD206 (5′-CATGAGGCTTCTCCTGCTTCTG-3; 5′-TTGCCGTCTGAACTGAGATGG-3), GAPDH (5′-ACGGCCGCATCTTCTTGTGCA-3′; 5′-CGGCCAAATCCGTTCACACCGA-3′).

### 2.11. ELISA Assay

The levels of the chemokines CXCL1 and CCL2, and cytokines TNF-α, IL-6, IL-10 and TGF-β were measured in the supernatants obtained from cell cultures or pleural cavity washes by ELISA, using commercially available antibodies according to the procedures supplied by the manufacturer (R&D Systems, Minneapolis, MN, USA).

### 2.12. Statistical Analysis

The data were analyzed by one-way analysis of variance (One-way ANOVA), followed by the Tukey test. When only two groups were evaluated, a Student t test was used. A value of *p* < 0.05 was considered significant. The results were presented as mean ± SEM (standard error of the mean). Calculations were performed using the GraphPadPrism 7.0 software for Windows (GraphPad Software, La Jolla, CA, USA).

## 3. Results

### 3.1. Db-cAMP Promotes Nonphlogistic Monocyte Recruitment to the Pleural Cavity of Mice Associated with Increased CCR2 Expression and CCL2 Levels and Depend on CCR2 and PKA

To investigate the ability of cAMP to induce leukocyte recruitment in vivo, db-cAMP was injected into the pleural cavity of BALB/c mice and the cells were collected at different time points. Db-cAMP induced a time-dependent influx of leukocytes into the pleural cavity, with increased cell numbers observed at 24 and 48 h time points (Figure 1A). The cells recruited to the pleural cavity were almost entirely mononuclear cells without any significant modification in neutrophil numbers. Indeed, db-cAMP-induced cell recruitment occurred in a nonphlogistic manner with an early (4 h) increase of IL-10 levels that persisted for up to 48 h, with no changes in the levels of the neutrophil chemoattractant CXCL1 and the pro-inflammatory cytokines TNF-α and IL-6 levels (Figure 1B).

To identify cell populations that migrated to the cavity, we analyzed the recruited cells at 48 h after db-cAMP injection for surface expression of F4/80, CD3, Ly6G and Ly6C. Our results show that injection of db-cAMP increases monocyte (Ly6C^+^ F4/80^−^) recruitment (Figure 1C,D), without changing the frequency of macrophages (F4/80^+^). Confirming the pleural counts of the cytospin slides (showed in Figure 1A), flow cytometry analysis showed that db-cAMP injection did not increase the frequency of neutrophils (Ly6G^+^) or lymphocytes (CD3^+^) in the pleural cavity when compared with PBS-injected mice (Figure 1D). Interestingly, there was increased frequency of CCR2^+^ monocytes (Figure 1E) associated with an early increased levels of CCL2 at 4 h time point (Figure 1F), whose levels remained until 48 h after db-cAMP injection (PBS: 8.6 ± 1.8; db-cAMP: 50.0 ± 18.1; CCL2 levels in pg/mL, *p* < 0.05, *n* = 5). The involvement of CCL2/CCR2 axis, an important determinant for mononuclear cell migration [53,54] in db-cAMP-induced migration was also evaluated by pharmacological inhibition of CCR2. Mice were pretreated with the antagonist of CCR2, RS504393 (2 mg/kg i.pl.), 1h before db-cAMP and we can observe that receptor inhibition decreased the db-cAMP-induced recruitment of mononuclear cells (Figure 1G).

In addition, we have investigated whether PKA (a cAMP effector protein) was involved in db-cAMP-induced mononuclear cells recruitment. For that, mice were pretreated with H89 (4 mg/kg, i.pl.), a PKA inhibitor, 1 h before db-cAMP and we can observe that PKA inhibition decreased the db-cAMP-induced recruitment of mononuclear cells (Figure 1H).

### 3.2. Db-cAMP Induces Macrophage Polarization towards M2 Phenotype in a PKA Dependent Manner

Next we examined the effect of db-cAMP on macrophage reprogramming, a key step for resolution of inflammation [10]. For this, bone marrow-derived macrophages (BMDMs) were treated with db-cAMP (100 µM) for different times (6, 12 and 24 h) and important markers of macrophage phenotypes M1 or M2 were evaluated.

As shown in Figure 2, db-cAMP increased the M2 macrophage markers Arg-1 (Figure 2A), CD206 (Figure 2B) and Ym-1 protein (Appendix A) in different expression kinetics. Db-cAMP also induced low expression of the M1 marker iNOS (Figure 2C), suggesting that macrophage phenotype induced by db-cAMP has a mixed profile. In addition, db-cAMP induced increased levels of IL-10 (Figure 2D) at 12 h, following the Arg-1 expression peak, without modifying TGF-β (Figure 2E) and TNF-α levels (Figure 2F). Noteworthy, db-cAMP showed similar effects on RAW 246.7 cells, by inducing the expression of Arg-1 (Appendix A) and CD206 (Appendix A), which peak at 24 h, and augmented iNOS levels (Appendix A).

To investigate if db-cAMP induces polarization of macrophages toward an M2 phenotype in a manner dependent of PKA, the main cAMP effector protein, BMDMs were pretreated for 1 h with the PKA inhibitor H89 (20 µM), and stimulated with db-cAMP (100 µM) for further 6 h. Interestingly, pre-treatment with H89 inhibited Arg-1(Figure 2G) and CD206 (Figure 2H) expression, and IL-10 (Figure 2I) levels in response to db-cAMP, suggesting the participation of PKA in db-cAMP-induced macrophage polarization.

### 3.3. Db-cAMP has Synergic Effect with IL-4 to Induce Phosphorylation of STAT3 and Arg-1 Expression

To investigate the intracellular signaling pathways triggered during db-cAMP-induced M2 polarization, we initially evaluated whether db-cAMP would to induce activation of STAT3 and STAT6, two important pathways for IL-4-induced M2 polarization [55]. For this, BMDMs were treated for different times (5, 10 and 30 min) with db-cAMP (100 µM) and phosphorylation of STATs 3 and 6 were analyzed by western blot. Figure 3A shows that db-cAMP promoted STAT3 phosphorylation within 30 min after treatment. Under these same experimental conditions, it was not possible to detect pSTAT6 levels (data not shown). Then, we evaluated whether db-cAMP would have synergistic effect with IL-4 on STAT3 phosphorylation. For this, BMDMs were treated with IL-4 at low concentration (5 ng/mL), db-cAMP (100 µM) or db-cAMP (100 µM) + IL-4 (5 ng/mL). Interestingly, an additive effect of db-cAMP with IL-4 was observed on STAT3 phosphorylation (Figure 3B).

Next, the effect of db-cAMP plus IL-4 on the expression of the M2 marker Arg-1 was evaluated. BMDMs were either treated with db-cAMP (100 µM) or IL-4 (20 ng/mL) alone or db-cAMP (100 µM) + IL-4 (20 ng/mL) for 6 h. Analysis of Arg-1 expression by qPCR also demonstrated a synergistic effect of db-cAMP with IL-4 (Figure 3C), showing increased Arg-1 expression when compared to db-cAMP or IL-4 stimuli alone.

### 3.4. Db-cAMP Re-educates M1 Macrophages toward an M2-like Phenotype Associated with Inhibition of STAT1 Phosphorylation

Having established the effect of db-cAMP on M2 polarization, we evaluated its impact on M1 polarization induced by pro-inflammatory stimuli.

BMDMs were stimulated with db-cAMP (100 µM) alone, LPS (10 ng/mL) + IFN-γ (10 ng/mL) or pretreated with db-cAMP (100 µM) for 1 h and then stimulated with LPS (10 ng/mL) + IFN-γ (10 ng/mL) for further 6 h. Db-cAMP was able to induce Arg-1 expression (Figure 4A) even in the presence of the M1 stimuli (LPS + IFN-γ), without changing iNOS expression (Figure 4B). In addition, the pre-treatment with db-cAMP prevented the production of the pro-inflammatory cytokines TNF-α (Figure 4C) and IL-6 (Figure 4D), and STAT1 phosphorylation (Figure 4E), which participates in M1 signaling pathway induced by LPS + IFN-γ [55].

### 3.5. Treatment of LPS-Inflamed Mice with db-cAMP Decreased the Numbers of M1 Macrophages without Modifying M2 and Mres, and Increases the Engulfment-Related Molecules AnxA1 and CD36

We next examined the role of db-cAMP in macrophage polarization in vivo, using a well-established model of acute inflammation—a self-resolving model of pleurisy. In this model, M1-like macrophages are predominant in the pleural cavity within 8 h after LPS intrapleural injection (peak of inflammation), while M2-like macrophages predominates during the resolution-phase of inflammation (characterized by clearance of the infiltrated neutrophils) [56]. In the present work, BALB/c mice were injected with LPS (250 ng/cavity i.pl.), treated at the peak of inflammation (8 h) with db-cAMP (4 mg/kg, i.pl.), and macrophages were characterized by flow cytometry 30 h after LPS challenge. 

The evaluated time (30 h) was chosen because this time point precedes the resolution phase, so it is predicted still have a mixed macrophage population with M1 and M2 phenotypes into the pleural cavity. Macrophage subpopulations were classified as M1 [F4/80^low^ GR1^+^ CD11b^med^], M2 [F4/80^high^ GR1^−^ CD11b^high^] or Mres [F4/80^med^ CD11b^low^] as previously published [12,44,45,46]. Analysis of cells obtained from pleural lavage showed that db-cAMP decreased the number of macrophages M1 (Figure 5A), but did not change the number of M2 (Figure 5B) and Mres (Figure 5C) macrophages. Representative dot plots and histograms are shown in Appendix A. Interestingly, in this same model, db-cAMP increased the frequency of macrophages (F4/80^+^) positive for AnxA1 (Figure 5D) and CD36 (Figure 5E), in which are molecules involved in phagocytosis of apoptotic neutrophils, a process called efferocytosis [57,58,59,60,61,62,63,64,65].

### 3.6. Db-cAMP Enhances Efferocytosis of Apoptotic Neutrophils in a PKA Dependent Manner

As db-cAMP promotes macrophage polarization to a M2-like phenotype and increases engulfment-associated molecules, all related to a pro-efferocytic macrophage phenotype [66], we next investigated whether db-cAMP plays a role in efferocytosis using previously described in vitro and in vivo efferocytosis assays [46]. Briefly, BMDMs were co-cultured with CFSE-labeled apoptotic human neutrophils (1:3, respectively) for 1 h. Then, the cells were washed and labeled with F4/80 (macrophage marker) and analyzed by flow cytometry. Representative histograms are shown in Figure 6A. Engulfment of apoptotic neutrophils by BMDMs treated with db-cAMP were increased compared to untreated macrophages, as indicated by the higher MFI of CFSE-labeled neutrophils in F4/80^+^ cells in this group (Figure 6B). This effect was partially reduced by pretreatment of cells with H89 (Figure 6C). Next, we evaluated in vivo efferocytosis in a peritonitis model, as described [37,45,46,50,67]. BALB/c mice received an intraperitoneal injection of zymosan (0.1 mg/i.p.) to induce macrophage recruitment. In this model of transient peritonitis, PMN cells peak at around 12 h (onset) followed by their clearance, and after 24 h there are predominance of macrophages, but still a residual of neutrophils and eosinophils [49]. After 62 h from zymosan injection, mice were treated with db-cAMP (4 mg/kg, i.p.) and 7 h later were injected intraperitoneally with CFSE-labeled apoptotic neutrophils. Cells were collected from the peritoneal exudate 1 h later, labeled with F4/80 and analyzed by flow cytometry and light microscopy to check for efferocytosis. Db-cAMP was able to induce increased efferocytosis, as observed by % of F4/80^+^/CFSE^+^ cells and MFI CFSE in F4/80^+^ ascertained by flow cytometry (Figure 6D–F), and morphological analysis by light microscopy (Figure 6G,H).

### 3.7. Inhibition of the cAMP Pathway with a PKA Antagonist H89 Prevents Natural Resolution of Pleurisy and Decreases the Numbers of Mononuclear Cells

We have previously shown that cAMP elevating compounds drive resolution of neutrophilic [32,34] and eosinophilic [33] inflammation by promoting apoptosis of these PMN cells. 

In the present study, we gathered data showing that cAMP also has effects on other key steps of resolution: macrophage reprogramming and efferocytosis. Therefore, our next step was to evaluate the effect of endogenous cAMP in physiological resolution of inflammation. For that, we used a well-established self-resolving model of LPS-induced pleurisy, in which the intrapleural injection of LPS induces a time-dependent influx of leukocytes into the pleural cavity, characterized by early peak of neutrophil (8 h after LPS injection), and a late incoming of mononuclear cells (24 h after LPS injection) [34,44,46,48]. Therefore, we injected the PKA inhibitor H89 at the peak of inflammation and recovered the cells 24 h after, for counting of neutrophils and mononuclear cells. As can be seen, the inhibition of PKA with H89 increases the number of neutrophils (Figure 7A) and decreases the number of mononuclear cells (Figure 7B) when compared to the group of mice that did not receive the inhibitor. We also performed experiments aiming to calculate the resolution indices [47]. Mice were challenged with LPS and injected with H89 at 8 h and 24 h (booster dose). The cells were collected 8 and 48 h after LPS injection for evaluation of neutrophils numbers and quantification of the resolution interval (*Ri*) [68,69]. Our data show that in this self-resolving model of inflammation the resolution interval (*Ri*) was ~30 h, whereas H89-treated animals remains with higher numbers of neutrophils into the pleural cavity and did not resolve, even at the last time point evaluated, with a predicted *Ri* > 48 h (Figure 7C). This data suggest that endogenous cAMP is crucial for resolves inflammation in the LPS-induced pleurisy model.

## 4. Discussion

Cyclic AMP is a second messenger that has multiple roles in cell physiology and exerts general modulating effects on a variety of cells. In the immune system, cAMP regulates the activities of innate and adaptive immune cells [27], showing the importance of cAMP in modulating inflammation. In the current study, by using db-cAMP (a cell permeable cAMP mimetic) we have shown that db-cAMP (i) induces nonphlogistic monocyte recruitment in vivo (depending on PKA and CCR2 signaling); (ii) induces macrophage polarization toward an M2 anti-inflammatory phenotype in vitro (depending on PKA) and has a synergistic effect with IL-4 on macrophage polarization to M2 phenotype; (iii) re-educates macrophages induced by IFN-γ/LPS toward an M2-like phenotype; (iv) decreases the number of M1 macrophages induced by LPS-injection and increases the frequency of cells expressing molecules involved with recognition of apoptotic cells and engulfment; (v) promotes efferocytosis of apoptotic neutrophils in vitro (depending on PKA) and in vivo; and that (vi) cAMP is an endogenous determinant of the physiological resolution of LPS-induced pleurisy. Therefore, we provide evidence that cAMP regulate key macrophage functions by regulating nonphlogistic monocyte migration, reprogramming macrophages, and promoting efferocytosis (Figure 8).

Recruitment of monocytes that differentiate into macrophages at inflammatory sites is a hallmark of inflammation resolution [3,70]. These cells promote clearance of the neutrophilic infiltrate, and therefore rescue apoptotic neutrophils from evolving to a necrotic state, what would contribute to amplify the pro-inflammatory process [8]. A recent study showed that rolipram, a PDE4 inhibitor, which increases the intracellular cAMP levels, induces chemotaxis of macrophages previously stimulated with LPS [71]. Here, we demonstrated that db-cAMP by itself was able to induce nonphlogistic monocyte migration, which may contribute to the pro-resolving properties of this molecule (Figure 7 and references [32,33,34]). Importantly db-cAMP-induced leukocyte migration was selective for monocytes, without causing neutrophil recruitment or pro-inflammatory cytokine/chemokine production, at least at the time points evaluated herein. In addition, mononuclear cell recruitment was accompanied by increased levels of the anti-inflammatory cytokine IL-10. Indeed, molecules that play proresolving actions such as AnxA1 [39,40,72], lipoxin A4 [73,74] and plasminogen/plasmin [46,75] are able to induce nonphlogistic macrophage recruitment contributing to the resolution of inflammation.

CCR2 receptor is involved in both migration of bone marrow monocytes into the bloodstream [53] and in the migration of mononuclear cells from blood to tissues [76]. Several cell types, including endothelial, mesothelial, fibroblasts, epithelial, smooth muscle, mesangial, astrocytes, monocytes/macrophages, and microglial cells [77,78,79,80,81], produce CCR2 receptor ligand CCL2 chemokine. CCL2, in turn, is responsible to regulating monocyte recruitment and activation of T lymphocyte subpopulations [82,83]. Indeed, Cailhier et al. have suggested that mesothelial cells are responsible for the production of this chemokine in the model of carrageenan-induced pleurisy [84]. Here, we have shown that db-cAMP induces early CCL2 production into the pleural cavity, which precedes monocyte recruitment. Thus, our hypothesis is that the detected CCL2 is produced by resident pleural cells, as in the study of Cailhier et al. [84]. By contrast, Motoyoshi and colleagues demonstrated that forskolin, an adenylyl cyclase inducer, decreased CCL2 levels induced by S100B, a RAGE receptor-binding protein involved on development of autoimmune and inflammatory diseases [85]. However, to the best of our knowledge, the present study is the first to demonstrate the participation of cAMP in CCR2/CCL2 pathway for monocyte recruitment in a non-inflammatory context. Together, the results of our work and the previous study [85] point to a dual role of cAMP in CCL2 production. In a phlogistic context of an ongoing inflammatory process, as shown by Motoyoshi, where there is an initial stimulus for migration of inflammatory monocytes/macrophages, increased cAMP levels may act reducing the CCL2 levels and limiting the inflammatory process [85]. One the other hand, cAMP may activate the CCL2/CCR2 axis, such as during the resolving phase of inflammation, and induce nonphlogistic monocyte recruitment, which may contribute to the resolution process. Thus, the effect of cAMP on monocyte migration seems to depend on the phase of inflammation, having divergent action at the onset and resolving phases of inflammation. In both cases, the outcome is the modulation of the inflammatory process favoring the resolution of inflammation.

Macrophages are plastic cells that differ according to the microenvironment and may assume different phenotypes in vivo and in vitro depending on the stimulus. Macrophage phenotypes are generally divided in M1 and M2 (M2a, M2b and M2c), also known as classically and alternatively activated macrophages, respectively. IFN-γ, LPS and TNF induce M1-like pro-inflammatory phenotypes, whereas the M2-like anti-inflammatory phenotype is induced by IL4/IL13 (M2a), immune complexes (M2b) and IL-10 or TGF-β (M2c) [86,87,88,89]. The production of cAMP by resolution phase macrophages (rMs), but not chronic inflammatory macrophages, was introduced by Bystrom and colleagues in 2008. Db-cAMP was able to transform M1-like macrophages to rMs-like cells [29]. Later, another study demonstrated that 8-Br-cAMP, another cAMP analogue, induces Arg-1 expression in RAW 264.7 murine macrophages, being the first to suggest the direct participation of cAMP in macrophage polarization [31]. Here, we have shown that db-cAMP induced not only Arg-1, but also other M2 markers such as CD206, Ym-1 and IL-10 in BMDMs and RAW 264.7 cells, demonstrating more robustly that, in fact, cAMP is involved in macrophage polarization to an M2-like phenotype. Corroborating with our data, a recent study demonstrated in the experimental autoimmune encephalitis (EAE) model that forskolin increases M2 (Arg1, Mrc1, Fizz1 and Ym1) and decreases M1 (NOS2 and CD86) markers dependent on the ERK signaling pathway [30]. However, it is important to note that in our experimental settings db-cAMP also induced iNOS expression at low range as compared to IFN-γ/LPS (compare Figure 2C and Figure 4B), without impacting TNF-α levels, suggesting that db-cAMP and forskolin reprogram macrophages towards different profiles. The expression of iNOS is a mechanism associated with bacterial clearance, through the production of nitric oxide, a free radical that has bactericidal capacity [90]. Therefore, the mixed profile of macrophages induced by db-cAMP in the present study, composed by macrophages with both profiles, containing anti-inflammatory markers and the pro-inflammatory marker (iNOs), the latter known for its bactericidal capacity, might be important for the resolution of infectious inflammation. Indeed, our group demonstrated that PDE4 inhibition by rolipram increases bacterial phagocytosis during the late phase of infection [91], thus corroborating our hypothesis.

Signal transducers and activators of transcription proteins (STATs) transmit signaling of many inducers of the M1 and M2 macrophage phenotypes. IFN-γ/LPS treatment induces macrophage polarization to the M1-like phenotype via STAT1, while IL-4 induces macrophage polarization to the M2-like phenotype via STAT6 (canonical pathway) and STAT3 (non-canonical pathway) [92]. Sheldon and colleagues showed that 8-Br-cAMP has a synergistic effect with IL-4 on Arg-1 and STAT6 expression [31]. Interestingly, here we demonstrated that db-cAMP induced transient STAT3 phosphorylation and had a synergistic effect with IL-4 on STAT3 phosphorylation and Arg-1 expression. Although in our experimental conditions changes in p-STAT6 levels were not detected after db-cAMP stimulation, we cannot exclude the participation of STAT6 in this process. Corroborating our data, Koscso and colleagues demonstrated that adenosine, which binds to A2A and A2B receptors and stimulate the adenylyl cyclase to produce cAMP, has synergistic effect with IL-10 in inducing macrophage polarization towards an M2c profile in a STAT3-dependent manner [93]. Moreover, recent studies have argued that STAT3 activation can occur via PKA [94,95]. Although we have not investigated here whether db-cAMP-induced STAT3 activation was via PKA, we can hypothesize that PKA activation may be occurring in our settings of macrophage reprogramming.

Importantly, a study demonstrated that prostaglandin E2 (PGE2), a cAMP elevator, activates PKA, which inhibits SIK2 (a salt-inducible kinase) and, in turn, prevents the phosphorylation of CREB-regulated transcription coactivator (CRTC3) [96]. This allows the dephosphorylation of SIK2 sites on CRTC3, resulting in the translation of the later to the nucleus [96]. Once inside the nucleus, CRTC3 can interact with CREB (cAMP response element-binding) to promote CREB-dependent gene transcription, which in macrophages includes the IL-10 gene involved in macrophage polarization to an anti-inflammatory (M2) profile [96,97]. Moreover, the clinically approved drugs bosutinib and dasatinib promote macrophage polarization, characterized by the production of high levels of IL-10 and very low levels of pro-inflammatory cytokines, by inhibiting SIKs through CRTC3 dephosphorylation and inducing CREB-dependent gene transcription [98]. Interestingly, pharmacological or genetic inhibition of SIKs also leads to CRTC3 dephosphorylation in macrophages with further CRTC3 translocation to the nucleus, where it acts as a co-factor for CREB-dependent gene transcription, including transcription of IL-10 and markers of regulatory-like macrophages (M2b), such as SPHK1, LIGHT and Arg-1 [99]. These studies provide important evidence of the role of PKA in reprogramming macrophages and corroborate our data that increased IL-10 levels induced by db-cAMP occur via PKA. Taking all these studies into account, we can suggest that the rapid db-cAMP-induced STAT3 phosphorylation found in our study (Figure 3A) might be occurring through cAMP→PKA→CREB activation, which can, in turn, increases IL-10 levels. Whether SIKs are involved in this process will require further investigation.

As the inflammatory response evolves, macrophages change their phenotype from M1 to M2 and then to Mres, a key process for resolving inflammation [3,10,100,101]. We have shown here that db-cAMP increased Arg-1 expression, reduced pro-inflammatory cytokine levels (TNF-α and IL-6) and STAT1 phosphorylation in BMDMs in the presence of M1 stimulus (IFN-γ/LPS), indicating that db-cAMP can skew the macrophage phenotype induced by IFN-γ/LPS to the M2 phenotype. Corroborating with our in vitro data, db-cAMP also decreased the number of LPS-induced M1 macrophages in vivo without affecting the number of M2 and Mres macrophages. In an experimental model of autoimmune encephalomyelitis (EAE), an inflammatory disease of the central nervous system where both M1 (IFN-γ) and M2 (IL-4) cytokines are present, forskolin deviates this balance favoring the M2 phenotype [30], suggesting that cAMP-induced polarization might depend on the microenvironment.

Efferocytosis is a hallmark for homeostasis and resolution of inflammation and is characterized by phagocytosis of apoptotic cells [102]. Apoptotic cell clearance has profound effects on immune and adaptive responses in inflamed tissues [66]. It has been shown that pro-resolving mediators that are known to increase cAMP levels, such as melanocortins, are inducers of efferocytosis [103]. In addition, increased levels of cAMP and PGE2, one of the major lipid mediator synthesized by macrophages during the inflammatory response, induced by lysophosphatidylserine (lysoPS) binding expressed on apoptotic neutrophils are involved in efferocytosis, signaling through increased activity of Rac1, dependent on PKA [104,105,106]. There are some key molecules well known to be involved with macrophage efferocytosis. AnxA1, a protein with pro-resolving properties, which was previously shown to be induced by cAMP, is produced by both macrophages and neutrophils, facilitating apoptotic neutrophil efferocytosis [107]. In addition, several studies have shown the participation of AnxA1 in efferocytosis [45,58,59,60,61,62]. Another molecule involved in macrophage mediated efferocytosis is the CD36 receptor, which recognizes lipid ligands present on the surface of apoptotic cells [57,108] contributing to the modulation of efferocytosis [57,63,64,65]. In the present work, we have identified that db-cAMP increases the percentage of cells expressing AnxA1 and CD36 during LPS-induced pleurisy model, adding to the understanding of the molecular events involved in cAMP-induced efferocytosis.

In accordance with the increased expression of engulfment-molecules in macrophages treated with db-cAMP, we found that db-cAMP could indeed promote efferocytosis, in vitro and in vivo. Mechanistically, we have shown in BMDMs that db-cAMP increases efferocytosis of apoptotic neutrophils in a PKA-dependent manner. Similarly, we demonstrated that db-cAMP was also able to increase efferocytosis in vivo. In contrast, Rossi and colleagues demonstrated that PGE2, PGD2, db-cAMP and 8-bromo-cAMP decreased macrophage-mediated neutrophil efferocytosis of human monocytes [109], but treatments were performed just for 15 min. Therefore, we suggest that a more prolonged treatment time seems to be essential for macrophage capacitation by increasing the expression of molecules involved on recognition and engulfment processes, as demonstrated by the increased AnxA1 and CD36 in our study.

The effects of db-cAMP on monocyte recruitment, polarization and efferocytosis were shown to occur in a PKA-dependent manner, as the PKA inhibitor H89 was able to inhibit db-cAMP-induced effects. PKA is the best known and most studied cAMP effector protein [26,27]. cAMP binds directly to PKA promoting rearrangements that favors enzymatic activity [26,27]. Among other actions, PKA mediates the inhibition of inflammatory mediator production in macrophages [110]. Indeed, inhibition of PKA by H89 limits the cAMP-mediated resolution of acute inflammation [33,34]. H89 was also able to inhibit db-cAMP and rolipram-induced effects such as increased AnxA1 expression in THP-1 cells and resolution of inflammation in the LPS-induced pleurisy model [32]. Here, akin with previous data obtained by administrating pharmacological compounds that raises the intracellular levels of cAMP, such as rolipram and Forskolin [34], we have shown that endogenous cAMP works as a branch to counter-regulate the inflammatory process, since its inhibition by using H89 hampered the spontaneous resolution of inflammation.

It has been reported that cAMP is an important intracellular molecule that conveys the signals of several pro-resolving mediators, such as RvD1, RvD2, melanocortin and AnxA1 [32,35,36,37,38]. The list of molecules bearing pro-resolving actions is continuously growing, as recently described for plasmin [45,46,75] and IFN-β [111], well known molecules only recently shown to affect several steps of resolution, such as monocyte migration, macrophage reprogramming, efferocytosis and neutrophil apoptosis. Whether db-cAMP function as an intracellular mediator or an inductor of these newly described pro-resolving molecules deserves further investigation.

In conclusion, the present study adds contribution to the current knowledge of the effects of cAMP on resolution of inflammation [32,33,34], showing that db-cAMP induces proresolving properties in macrophages (summarized in Figure 8), such as nonphlogistic recruitment, reprogramming and efferocytosis, which are key processes for resolving inflammation.

## Figures and Tables

**Figure 1 cells-09-00128-f001:**
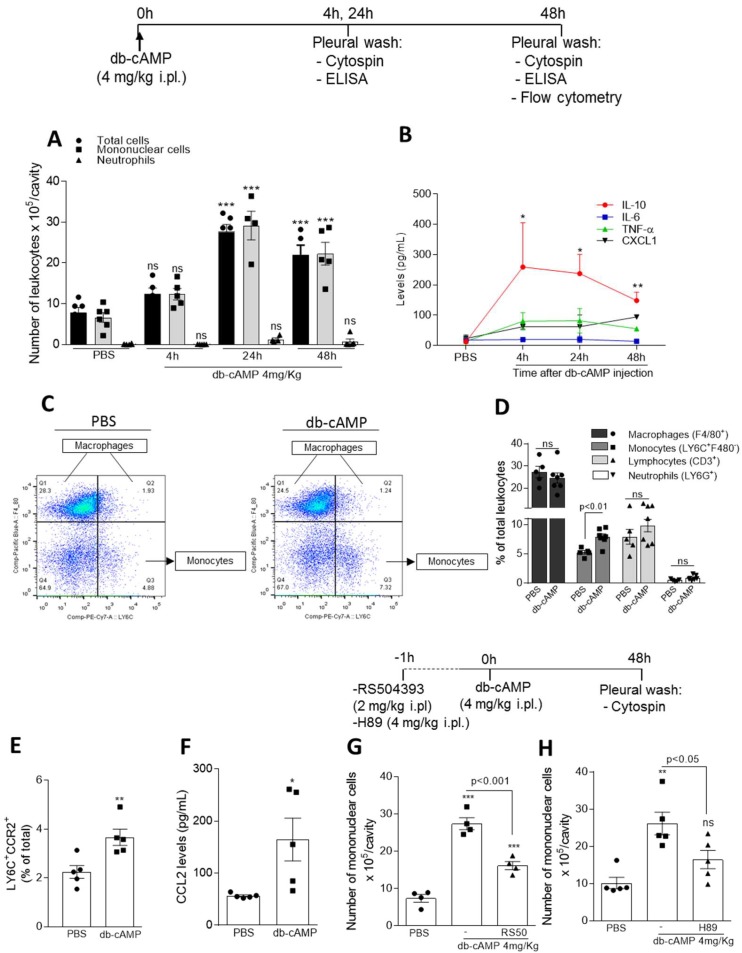
Time-course of leukocyte recruitment to the pleural cavity of mice after db-cAMP injection and effect of CCR2 and PKA inhibition. BALB/c mice were challenged by an i.pl. injection of db-cAMP (4 mg/kg) and the cells present in the pleural cavity were harvested at 4, 24 and 48 h. Pleural cells were processed for total and differential leukocyte counts of cytospin preparations by light microscopy (**A**). Levels of cytokines and chemokine (in pg/mL) (**B**) were measured by ELISA assay in the supernatants obtained from pleural cavity washes after PBS or db-cAMP injection. Flow cytometry analysis of pleural leukocytes collected 48 h after db-cAMP (4 mg/kg) or PBS injection (**C**,**D**). Representative dot plots (**C**) and percentage of lymphocytes, macrophages, monocytes, neutrophils (from the total leukocytes) (**D**), and monocytes CCR2^+^ (**E**). Levels of the chemokine CCL2 (in pg/mL) were measured by ELISA assay in supernatants obtained from pleural cavity washes 4h after db-cAMP injection (**F**). Mice were also pre-treated for 1 h with RS504393 (2 mg/kg, i.pl.) (**G**) or H89 (4 mg/kg, i.pl.) (**H**) and cells from pleural cavity were harvested 48 h after db-cAMP injection for counting of mononuclear cells. Results are expressed as the number of cells per cavity and are shown as the means ± SEM of 4–6 mice in each group. * *p* < 0.05, ** *p* < 0.01, *** *p* < 0.001 when compared with PBS-injected mice.

**Figure 2 cells-09-00128-f002:**
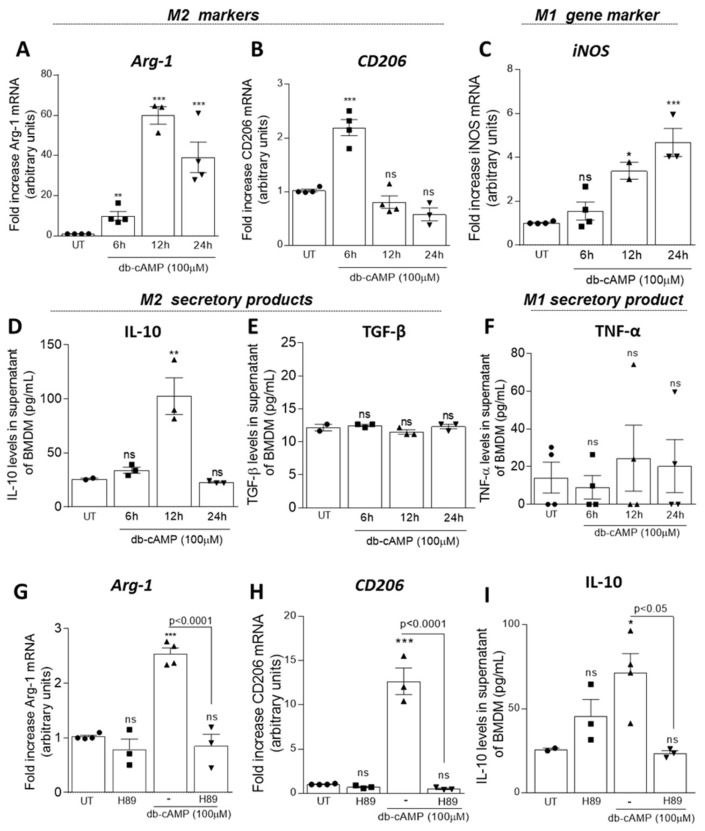
Effect of db-cAMP on macrophage polarization. Bone marrow from BALB/c mice was isolated and differentiated to bone-marrow-derived macrophages (BMDMs) for 7 days. Then BMDMs were treated with db-cAMP (100µM) for 6, 12 and 24 h and analyzed by qPCR for the expression of the M2 markers: Arginase-1 (**A**) and CD206 (**B**) and the M1 marker iNOS (**C**). The levels of secretory products of M2 macrophages TGF-β and IL-10 (**D**,**E**) and M1 macrophages TNF-α (**F**) were measured from supernatants by ELISA assay. In (**G**–**I**) BMDMs were pretreated with H89 (20 µM) for 1 h and then stimulated with db-cAMP (100 µM) for further 6 h to measurement of the M2 markers Arginase-1, CD206 and IL-10. Results are expressed as fold increase (qPCR) and levels in pg/mL (ELISA), and are shown as the mean ± SEM. * *p* < 0.05, ** *p* < 0.01, *** *p* < 0.001 when comparing with untreated (UT) BMDMs.

**Figure 3 cells-09-00128-f003:**
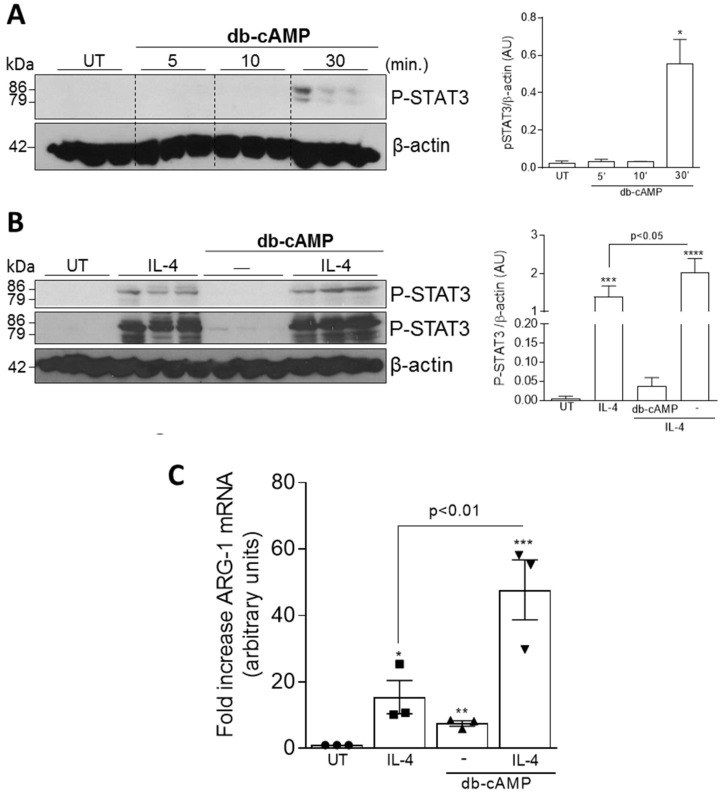
Effect of db-cAMP alone or in combination with IL-4 on STAT3 signaling and *Arginase-1* expression. BMDMs from BALB/c mice were treated with db-cAMP (100 µM) for 5, 10 and 30 min (**A**) or were stimulated with IL-4 (5 ng/mL), db-cAMP (100 µM) or db-cAMP (100 µM) + IL-4 (5 ng/mL) for 30 min (**B**). Cell lysates were subjected to western blot analysis to assess pSTAT3 (**A**,**B**). In **B**, two exposures of p-STAT3 are presented to show the full range of expression. β-actin was used as a loading control. Densitometry analyses are shown (**A**,**B**). In (**C**) BMDMs were untreated or stimulated with db-cAMP (100 µM), IL-4 (20 ng/mL) or db-cAMP (100 µM) + IL-4 (20 ng/mL) for 6 h, and analyzed by qPCR for the expression of Arginase-1. Results are expressed as fold increase (qPCR) and are shown as the mean ± SEM. * *p* < 0.05, ** *p* < 0.01, *** *p* < 0.001, **** *p* < 0.0001 when comparing with untreated (UT) BMDMs.

**Figure 4 cells-09-00128-f004:**
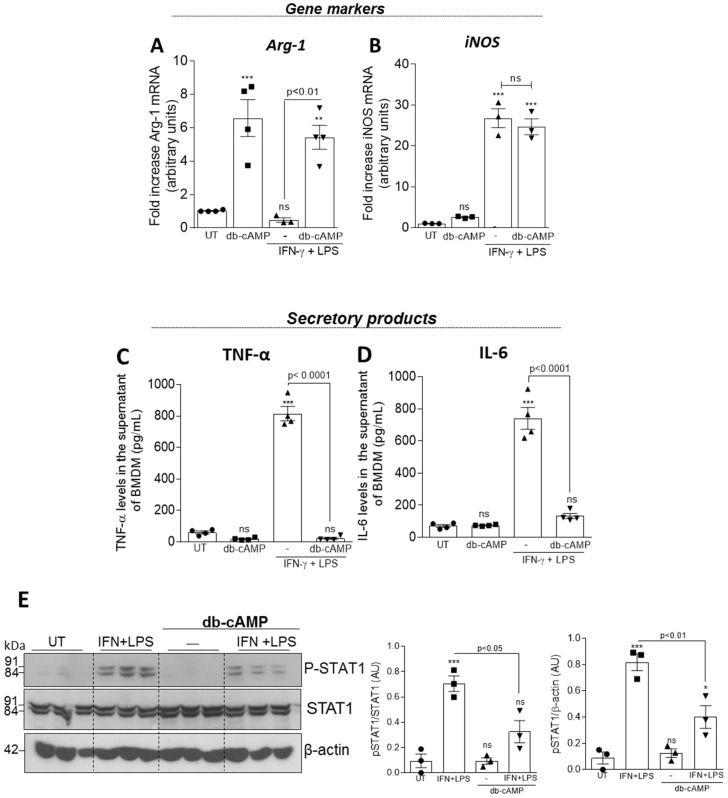
Effect of db-cAMP on macrophage polarization and STAT1 activation induced by IFN/LPS. BMDMs from BALB/c mice were stimulated with db-cAMP (100 µM), LPS (10 ng/mL) + IFN-γ (10 ng/mL) or were pretreated for 1 h with db-cAMP (100 µM) and then stimulated with LPS (10 ng/mL) + IFN-γ (10 ng/mL) for further 6 h. Cells were collected for measurement of the M2 marker Arg-1 (**A**) and the M1 marker iNOS (**B**) by qPCR, and the culture supernatants were used for measurements of the secretory products of M1 - TNF-α (**C**) and IL-6 (**D**) by ELISA. In (**E**) BMDMs were stimulated with LPS (10 ng/mL) + IFN-γ (10 ng/mL), db-cAMP (100 µM), or were pretreated with db-cAMP (100 µM) for 1 h and then stimulated LPS (10 ng/mL) + IFN-γ (10 ng/mL) for 30 min. Cell lysates were subjected to western blot analysis to assess the phosphorylated (pSTAT1) and total STAT1 levels (**E**). qPCR, ELISA assay and WB densitometry are expressed as fold increase, levels in pg/mL and arbitrary units, respectively, and are shown as the mean ± SEM. * *p* < 0.05 ** *p* < 0.001, *** *p* < 0.0001 when comparing with untreated (UT) BMDMs.

**Figure 5 cells-09-00128-f005:**
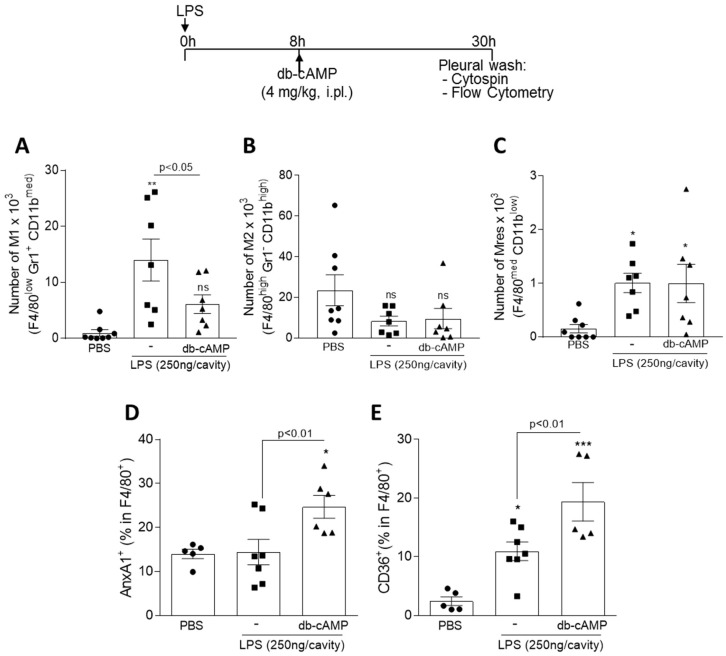
Effect of the treatment of LPS-inflamed mice with db-cAMP on macrophage polarization and engulfment-related molecules. BALB/c mice were injected with LPS (250 ng/cavity, i.pl.) or PBS, and 8 h later received an injection of db-cAMP (4 mg/kg, i.pl.). Cells present in the pleural cavity were harvested 30 h after LPS challenge or PBS injection. The number of M1 [F4/80^low^ GR1^+^ CD11b^med^] (**A**), M2 [F4/80^high^ GR1^-^ CD11b^high^] (**B**), Mres [F4/80^med^ CD11b^low^] macrophages (**C**) and % of AnxA1 and CD36 in F4/80^+^ cells (**D** and **E**) were evaluated by flow cytometry. Results are shown as the mean ± SEM of 5–8 mice in each group. * *p* < 0.05, ** *p* < 0.01, *** *p* < 0.0001, when compared with PBS-injected mice.

**Figure 6 cells-09-00128-f006:**
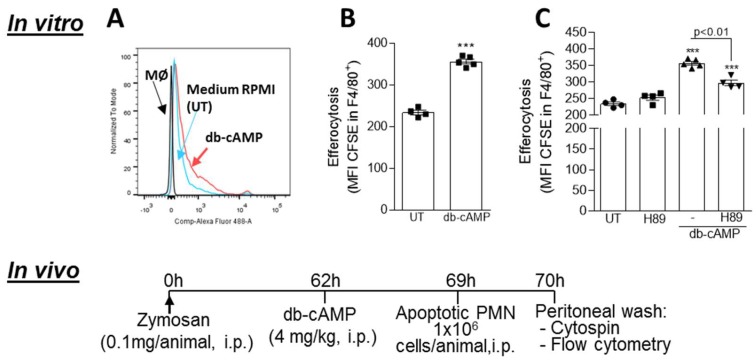
Effect of db-cAMP on macrophage efferocytosis. The in vitro efferocytosis assay was performed by co-culturing BMDMs from BALB/c mice with human apoptotic neutrophils labeled with CFSE in a proportion of 3 neutrophils per macrophage. Macrophages were treated with db-cAMP (100 µM) for 24 h (**B**), or pre-treated with H89 (20 µM) for 1 h prior to treatment with db-cAMP for further 24 h (**C**). Efferocytosis was assessed by flow cytometry and was expressed as MFI of CFSE-labeled neutrophils in F4/80^+^ cells. Representative histograms are shown in (**A**). To determine efferocytosis in vivo, BALB/c mice received an i.p. injection of 0.1 mg of zymosan and 62 h later were injected i.p. with db-cAMP (4 mg/kg) or PBS. Seven hours after the treatment, mice received an i.p. injection of 10^6^ apoptotic human neutrophils labeled with CFSE. The cells from the peritoneal cavity were collected 1 h later. Efferocytosis was assessed in macrophages by flow cytometry according to their size and granularity, expression of the surface molecule F4/80, and intracellular CFSE (**D**,**E**), as described in the gating strategy (**F**). Results are expressed as % of MØ F4/80^+^ CFSE^+^ (**D**) or MFI of CFSE-labeled neutrophils in F4/80^+^ cells (**E**) of 6–7 mice in each group. Efferocytosis was also investigated by morphological counting of cytospin slides stained with May-Grunwald-Giemsa (**G**,**H**). In **H**, arrows indicate apoptotic neutrophils inside macrophages. Magnification 40×. Results are shown as the mean ± SEM * *p* < 0.05, *** *p* < 0.001, when compared with PBS-injected mice.

**Figure 7 cells-09-00128-f007:**
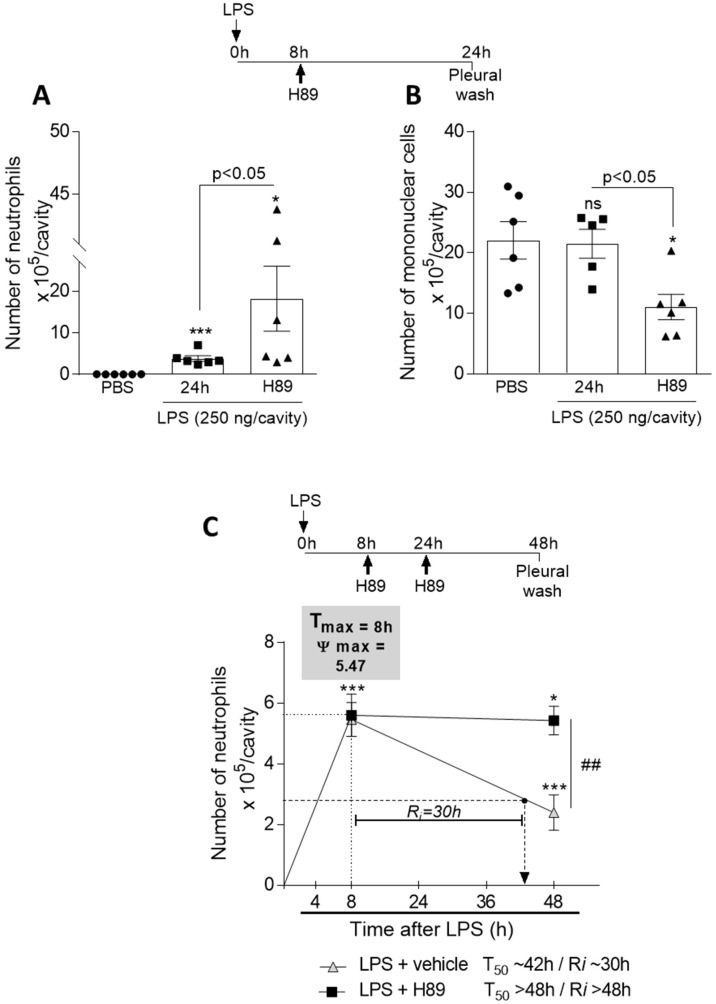
Effect of PKA inhibition in a self-resolving model of inflammation. BALB/c mice were injected with LPS (250 ng/cavity, i.pl.) or PBS, and 8 h later received an injection of H89 (4 mg/kg, i.pl.). Cells present in the pleural cavity were harvested 24 h after LPS challenge or PBS injection for neutrophil (**A**) and mononuclear cells (**B**) counts. In (**C**) mice were injected with LPS (250 ng/cavity, i.pl.) or PBS, and 8 and 24 h later received an injection of H89 (4 mg/kg, i.pl.). Cells present in the pleural cavity were harvested 8 h and 48 h after LPS challenge or PBS injection and neutrophils were counted from cytospin preparations to calculate resolution indices. Results are shown as the mean ± SEM of 5–7 mice in each group. * *p* < 0.05, *** *p* < 0.001, when compared with PBS-injected group.

**Figure 8 cells-09-00128-f008:**
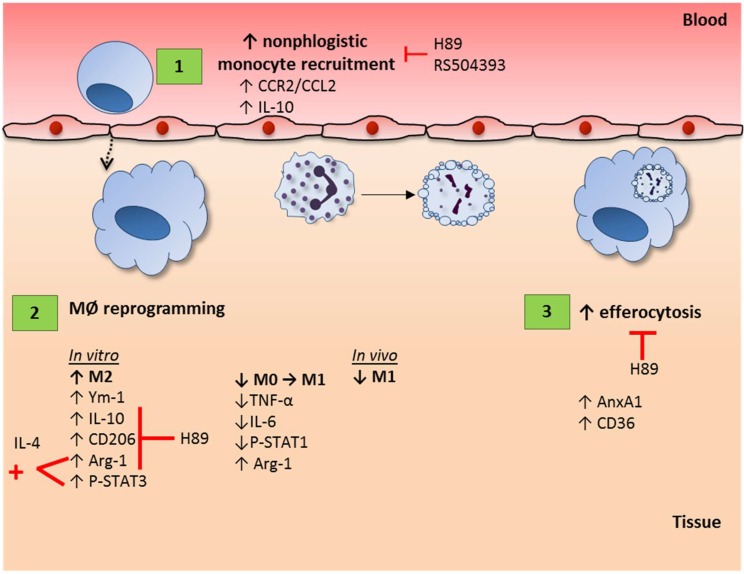
Schematic representation of the effects of cAMP on macrophage features. Our data show that cAMP induces monocyte recruitment in a PKA- andCCR2--dependent manner (1). We also demonstrate that cAMP promote macrophage polarization to the M2 profile in a PKA-dependent manner, while skew IFN-γ/LPS polarization to M2 phenotypes (2). Moreover, cAMP synergizes with IL-4 to promote M2 markers in macrophages. In addition, we demonstrate that cAMP promote the expression of engulfment-molecules and increases the efferocytic capacity of macrophages in a PKA-dependent manner (3). The pro-resolving effects summarized herein (1–3) have been described for db-cAMP, a cAMP mimetic, and can account for the endogenous role of cAMP in physiological resolution of inflammation. Note: Red lines represent inhibition effect.

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
