# Peer review of "Cyclic AMP Regulates Key Features of Macrophages via PKA: Recruitment, Reprogramming and Efferocytosis"

_cells, 2020, doi:10.3390/cells9010128_

Round 1
Reviewer 1 Report
In this manuscript, Negreiros-Lima and colleagues report the effects of db-cAMP on macrophage migration, gene expression, and efferocytosis. Overall, this well-written manuscript contains clear results reflecting strong experimental design. I have no doubt that db-cAMP modulates macrophage function in the manners reported in this manuscript. This work builds upon a large literature where the anti-inflammatory effects of cAMP/PKA signaling in macrophages have been well-documented over the past 20+ years. For this reason, the results here are largely anticipated and can be viewed as representing an incremental advance for this well-developed field. Moreover, db-cAMP is unlikely to be developed into a therapeutic agent. Beyond this perspective on the potential impact of this work, additional major and minor points requiring further clarification are listed below.
Major comments:
The authors note that protein kinase A is responsible for the modulatory capacity of db-cAMP in macrophages. However, little attention is paid to the downstream mediators of the effects of PKA. Recent work has clearly demonstrated that salt inducible kinases (SIKs) play a clear role in switching inflammatory gene expression patterns in macrophages. At the least, the authors should cite key studies in this rapidly-evolving field such as PMIDs 23033494, 23241891, 26590148, 25114223, 25351958, and 30150136 (review). The authors should explain the proposed pathway through which db-cAMP rapidly induces STAT3 phosphorylation in Figure 4A.
Minor comments:
The immunoblots in Figures 4A and 5E are lacking loading controls for the ‘total’ (unphosphorylated) forms of STAT3 and STAT1. In their discussion (page 16), the authors note reference 73 where it was demonstrated that forskolin reduces CCL2 levels. In Figure 2F, authors show that db-AMP increased CCL2. This potential discrepancy should be addressed in the discussion.Author Response
Please see the attachment

Reviewer 2 Report
The study by Negreiros-Lima et al describes a new role for cAMP in regulating important functions of monocytes/macrophages during the resolution of inflammation. The authors performed in vivo and in vitro studies in various animal models and macrophages cell types and claim to have determined that the cAMP analog db-cAMP promotes monocyte migration to tissues and macrophage reprogramming to a Th2-like phenotype as well as the uptake of apoptotic cells. However, the use of non-physiological systems and non-relevant macrophage types as well as significant flaws in logic, experimental design and data interpretation should be rectified to increase its merit and justify the suggested conclusions.
Major comments:
The results in Figure 1 does not make any sense. Is there any evidence that micromolar concentrations of cAMP are produced during the resolution of inflammation? If so, how do they attract monocytes? Is there a receptor for cAMP? If it is presumably through the activation of PKA and release of CCL2 by macrophages than this is not a chemotactic response but a chemokinetic or a repulsive response. Since the relevance of these line cells to resolution is scarce and the data is perplexing I suggest omitting these data. In figure 2 it makes more sense that tissue cells are producing CCL2 that attracts monocytes. However, the production of CCL2 in pleural cavities was measured at 48 hrs rather than at 4 hrs, and the effect of its antagonist and H89 was not evaluated. Moreover, to affect migration, ERK should be activated much earlier than at 48 hrs and in a much more consistent manner. It is more likely that the activation of ERK in tissue cells is responsible for the chemotactic effect through CCL2 production and therefore is not detectable in monocytes. These experiments should be performed. The physiological role of cAMP was not demonstrated in this study although the addition of a cell permeable agonist is claimed to promote different events in the resolution of inflammation. To directly demonstrate the involvement cAMP in physiological resolution, cAMP antagonists or H89 should be injected in a model of resolving inflammation, and their actions in limiting macrophage reprogramming and efferocytosis should be examined in vivo. A recent report indicated IFNb as a novel proresolving cytokine that affects monocyte migration, reprogramming and efferocytosis and promotes human neutrophil apoptosis through STAT3, but not STAT1. It will significantly increase the merit of the current manuscript if the authors will examine IFNb expression in resolutive macrophages following cAMP supplementation and/or antagonism, and in vitro, with neutralization of IFNb, determine the impact on macrophage polarization and efferocytosis. Such studies will provide a downstream mechanism for cAMP-mediated efferocytosis and macrophage reprogramming. The effect of H89 on CD206 and IL-10 should also be shown in Fig. 3. In Fig. 4A and 5E the blots for total STAT3/1 should be shown and statistical significance for STAT3 should be reached by more repeats. Also, the effect of cAMP could be due to low release of IL-4. This should be examined by ELISA. In Fig. 6, representative dot plots and gating strategy should be shown. Also, to show a role in the resolving phase of the response, the cell populations should be tracked following cAMP at 48-72 hrs. In Fig. 7, flow cytometry does not necessarily show increased efferocytosis. In 1 hr this probably reflects adhesion of apoptotic neutrophils to macrophages rather than phagocytosis. The targets should be labeled with a pH sensitive dye that will reflect phagosomal maturation around the targets. The histology also shows many neutrophils with live nucli (not condensed). Please indicate the percentage of apoptotic PMN and how it was detected. Moreover, the flow analysis shows gating of F4/80 negative cells rather than F4/80 positive ones. Please clarify. Since CCL2 was previously shown to promote efferocytosis (PMID: 20691665), the effect of CCR2 antagonism on efferocytosis in vivo should be examined as well. The Bystrom study (line 476) showed that cAMP is important in the generation of rMs rather than Mres. These macrophages were generated in a different system than chronic inflammatory macrophages and no evidence of reprogramming was provided in this study. Please correct the discussion accordingly.Minor comments:
The original description of Mres as CD11blow macrophages was in Schif-Zuck et al. 2011. Please cite accordingly on lines 57 and 124 . In line 62 the correct statement is "M1-like macrophages can be re-educated into the M2 and consequently to the Mres phenotypes by various effector molecules or the uptake of apoptotic cells. Conversion from M2 to M1 has also been reported". Appropriate referencing of ref 10, Fadok et al, 1998 and Voll et al 1997 should be added. In lines 155, 163, 428, 488 and Fig. 8 legend it should be corrected to IFNg. In line 212 it should be chemokines. Line 233-234- delete the and pathways (a single molecule does not constitute a pathway). In Fig. 2E-eror bar is missing. Fig. 3A,G Y axis should be Arg-1, 3F – should be TNFalpha. Correct to Western blot across the manuscript Ref 41 should be corrected. Line 521, change to phagocytosis of apoptotic cells.Author Response
Please see the attachment.

Round 2
Reviewer 2 Report
The revised version of the manuscript is significantly improved. However, some changes still need to be made to bring the manuscript to a publishable quality.
The results regarding the role of ERK in cAMP mediated monocyte recruitment are still not convincing. ERK activation was expected at earlier times then 4 hrs, and inhibition with U0126 in vivo was not performed to exclude a role for ERK in CCL2 production. Moreover, the increase in ERK phosphorylation is accompanied with a reduction in actin expression, suggesting transcriptional changes have taken place 48 hrs after PKA activation. The lack of blot for total ERK is glaring (Suppl Fig. 1) and as a result changes in the total protein levels of ERK could not be excluded. I recommend omitting these results from the results, discussion and figure 8. In Fig. 7 the symbols legend is incorrect. Please reverse the order. The results regarding the role of cAMP in all aspects of resolution (neutrophil apoptosis, efferocytosis and macrophage reprogramming) are really in line with increased production of IFNb following efferocytosis as was recently reported. This could be easily examined using RT-PCR on macrophages recovered from one of the resolving models + db-cAMP and treatment with H89 and RS50, separately. This can be done without purchasing additional reagents and will bring the current publication up to speed with recent literature. It will also validate the role of CCL2 in efferocytosis (as presented in the response letter) and whether IFNb is a key mediator of the cAMP+efferocytosis action. In lines 553-555 it should be stated "The production of cAMP by resolution phase macrophages, but not chronic inflammatory macrophages, was introduced by Bystrom in 2008 (rMs). Db-cAMP was able to transform M1-like macrophages to rMs-like cells". The analysis of the FACS data in fig. S4 is presented in a confusing manner and it is not clear which cells are included in the statistics. Also, although the percentage of Mres does not seem to change significantly after db-cAMP, it does look like the expression of CD11b is reduced in the whole population. Please examine if this is statistically significant and if so add this data.Author Response
Please see the attachment.
